# Research Advances in the Analysis of Estrogenic Endocrine Disrupting Compounds in Milk and Dairy Products

**DOI:** 10.3390/foods11193057

**Published:** 2022-10-01

**Authors:** Jia Chang, Jianhua Zhou, Mingyang Gao, Hongyan Zhang, Tian Wang

**Affiliations:** Shandong Provincial Key Laboratory of Animal Resistance Biology, Key Laboratory of Food Nutrition and Safety, College of Life Sciences, Shandong Normal University, Jinan 250014, China

**Keywords:** dairy products, estrogenic disrupting compounds, analysis, pretreatment

## Abstract

Milk and dairy products are sources of exposure to estrogenic endocrine disrupting compounds (e-EDCs). Estrogenic disruptors can accumulate in organisms through the food chain and may negatively affect ecosystems and organisms even at low concentrations. Therefore, the analysis of e-EDCs in dairy products is of practical significance. Continuous efforts have been made to establish effective methods to detect e-EDCs, using convenient sample pretreatments and simple steps. This review aims to summarize the recently reported pretreatment methods for estrogenic disruptors, such as solid-phase extraction (SPE) and liquid phase microextraction (LPME), determination methods including gas chromatography-mass spectrometry (GC-MS), liquid chromatography-mass spectrometry (LC-MS), Raman spectroscopy, and biosensors, to provide a reliable theoretical basis and operational method for e-EDC analysis in the future.

## 1. Introduction

According to the U.S. Environmental Protection Agency (EPA), endocrine-disrupting compounds (EDCs) are exogenous agents that can interfere with the synthesis, secretion, transport, metabolism, binding, or elimination of natural blood-derived hormones, which are responsible for homeostasis, reproduction, and development processes [1]. Estrogen receptors (ERs) are identified as key targets of EDCs. There are several natural and artificial compounds that can bind to the ERs to interfere with their activities, either act as agonists to activate biological reactions (e.g., genistein and bisphenol A (BPA)), or serve as antagonists that compete with endogenous hormones (e.g., 17β-Estradiol (17β-E2)) to inhibit the function of ERs [2]. EDCs have adverse effects on the development, reproduction, nerves, and immunity of humans and animals by interfering with the endocrine system [3]. Estrogenic EDCs (e-EDCs) are hormone estrogens or chemicals that can simulate or induce estrogenic reactions in organisms. They are hormonally active even at low concentrations and are a subclass of EDCs [4]. According to previous studies, e-EDCs may affect the human endocrine system by disrupting hormone synthesis, action, and metabolism. At present, the health risks associated with dietary exposure to e-EDCs have attracted increasing attention because of the adverse effects of these chemicals on reproductive and developmental disorders in the next generations. Their possible roles in human carcinogenesis have also been documented [5]. The development of sensitive and rapid approaches for the analysis of e-EDCs is the basis of risk assessment. In recent years, many reports have highlighted the application of different methods for e-EDCs analysis, focusing on either e-EDCs in the environment or a specific type of e-EDCs in food matrices. Milk and dairy products have been reported to be a source of e-EDCs [6]. They are also an important part of children’s diets. Therefore, it is important to detect the e-EDCs in milk and dairy products. In this study, the sample pretreatment and detection methods of different e-EDCs in milk and dairy products are reviewed for the first time to provide references for the establishment of more sensitive and accurate detection approaches for the quality management of milk and dairy products.

### 1.1. Introduction to e-EDCs

Different e-EDCs have been identified in milk and dairy products, including endogenous natural estrogens, synthetic estrogens, phytoestrogens, fungal estrogens, and other e-EDCs (Figure 1).

Endogenous natural estrogens in animals, also known as natural steroid estrogens, include estrone (E1), estradiol (17β-E2 and 17α-E2), estriol (E3), and their glucuronidation and sulfate metabolites. In cows, estrogens are mainly synthesized in the ovaries and placenta. Lipophilic estrogens can easily pass through the blood-milk barrier, making their concentrations in milk directly related to blood levels. Therefore, throughout pregnancy, the concentration of estrogens in milk increases with the levels of estrogens secreted by the placenta [7]. Moreover, the types of animal products, animal species, sex, age, and physiological status largely determine the concentration of natural steroid estrogens [8]. Studies have shown that through the risk quotient and optimized risk quotient method, the level of environmental risk is usually expressed as 17β-E2 > E1 > 17α-E2 > E3 [9]. Estrogenic activity is used to predict the risk; the stronger the activity, the greater the risk. Among the endogenous estrogens, E2 is a major natural estrogen with the greatest estrogenic activities. E1, a metabolite of E2, is a weak estrogen. E3 is the weakest natural estrogen, accounting for only 10% activity of E2 [10]. This is because E2 can react with two receptors (ERα and ERβ) simultaneously and has the highest binding affinity for ERα [11]. The Codex Alimentarius Commission (CAC) and the Chinese National Food Safety Standard stipulate that E2 cannot be detected in animal foods [12,13]. The European Union (EU) and the CAC propose that the maximum amount of exogenous E2 entering the human body through food should not exceed 50 ng·kg^−1^/day [14,15] (Table 1). E-EDCs with anabolic effects are synthetic estrogens, including dienestrol (DS), diethylstilbestrol (DES), BPA, hexestrol, (HEX), and 17α-ethynyl estradiol (17α-EE2) [5]. Dienestrol (DS), a stilbene derivative, is a catabolic DES. The estrogenic activity of DS is similar to that of the endogenous natural estrogen E2. Exposure to DES is associated with breast cancer in humans [16]. Since the 1980s, the EU has banned adding hormones to animal feed. The Ministry of Agriculture of China has also issued a notice to prohibit the addition of hormones, such as DES and E2, to animal feedstuffs and drinking water [17,18]. BPA is an environmental and food pollutant used as a food packaging additive. Its estrogenic activity is weak for the general population but toxic to infants [19]. The Food and Drug Administration (FDA) recommends that the maximum intake dose of BPA in the human body is 50 µg·kg^−1^/day [20], and the maximum residue limit (MRL) of BPA in foods stipulated by the EU is 0.05 mg·kg^−1^ [21] (Table 1).

Phytoestrogens belong to a large family of plant-derived nonsteroidal compounds that are structurally similar to E2. They can combine with ERs and exert weak estrogenic or anti-estrogenic effects. Many ruminant feeds are supplemented with isoflavones, coumarins, and lignans, which can be digested and excreted in milk [22]. Estrogen-active mycotoxins, such as zearalenone (ZEN), produced by *Fusarium* spp. are common contaminants in corn silage and other cereals. In ruminants, ZEN can be converted into α-zearalenol and β-zearalenol, which are bioactive derivatives. These fungal estrogens have been reported to be present in milk [23]. Zeranol derived from ZEN is banned as a growth promoter in cattle in the EU and China and cannot be detected in beef and other cattle food products [24] (Table 1).

**Table 1 foods-11-03057-t001:** Maximum Residue Limits (MRLs) and Daily Tolerable Intakes (TDIs) for some e-EDCs.

E-EDCs	MRL (s) or TDI (s)	Products	Provenances	Refs.
E2	Not Detected	Animal foods	The CAC and the Chinese National Food Safety Standard	[12,13]
E2	Not more than 50 ng·kg^−1^/day	Food intake	The EU and the CAC	[14,15]
DES and E2	Not Detected	Animal feedstuffs and drinking water	The EU and Ministry of Agriculture of China	[17,18]
BPA	50 µg·kg^−1^/day	Food intake	The FDA	[20]
BPA	0.05 mg·kg^−1^	Foods	The EU	[21]
Zeranol	Not Detected	Beef and other cattle food products	The EU and China	[24]

In addition, alkylphenols (e.g., 4-tert-octylphenol, 4-t-OP) and other chemicals such as polycyclic aromatic hydrocarbons (PAH), pesticides, polychlorinated biphenyls (PCBs), dichlorodiphenyltrichloroethane (DDT), some drugs (such as antiepileptic drugs), fungicides, and cotinine are considered to be e-EDCs [25]. The added 4-t-OP was used as a surfactant in the detergent, plastic, and pesticide formulations [26]. They are similar to natural estrogens and have a higher estrogenic activity than homogeneous alkylphenol phases [27].

### 1.2. Toxic Effects of e-EDCs

E-EDCs enter the human body through water and food intake, inhalation, and skin [28]. These chemicals affect steroidogenesis, folliculogenesis, and spermatogenesis and can also cause complications of pregnancy, genital malformations, and cancer and may lead to multigenerational and intergenerational effects [29,30]. If the mother (F0 generation) is exposed to e-EDCs during pregnancy and development, her offspring (F1 generation) are directly exposed to e-EDCs, and her grandchildren (F2 generation) develop directly from the exposed F1 generation germ cells. The direct effects of e-EDCs on F1 or F2 generations are known as multigenerational exposure [31]. However, when the father (generation F0) or non-pregnant women (generation F0) are exposed to e-EDCs, the effects on the F1 generation occur through germ cell exposure, which is a multigenerational exposure, whereas the effects on the F2 generation (the first generation without direct exposure) are referred to as transgenerational exposure in nature [32]. The intergenerational effects are thought to be caused by epigenetic mechanisms. Epigenetic changes in germline cells include DNA methylation, histone modification, and non-coding RNA. Epigenetic changes might be passed on to the unexposed generation through germlines, thus affecting the offspring and causing intergenerational phenomena [33]. E-EDCs have been confirmed to target a variety of hormonal systems and have adverse effects on the reproductive system, adolescence, embryonic development, and fetal sex differentiation [34]. Endogenous nature estrogen levels are quite low in children, and small amounts of e-EDCs can disrupt and destroy the development of the urogenital tract, mammary glands, and central nervous system [35]. Exposure to BPA can lead to the decreased fertility in mammals owing to premature activation of primordial follicles and altered levels of sexual steroids. Approximately 90% of the estrogens in the environment are produced from animal feces and sewage, which are rarely processed [36]. They are capable of disrupting aquatic ecosystems, resulting in the emergence of intersex fish [37], and the feminization of male fish [38]. Studies have shown that some e-EDCs, such as HEX, DES, and DS, may inhibit the hydrolytic activity of pancreatic lipase by blocking substrate binding to this key digestive enzyme [39].

### 1.3. Exposure Routes of e-EDCs

Unmetabolized e-EDCs or estrogen-active metabolites present in feces can be transferred from field feces to groundwater and finally to surface water [40]. In the environment, e-EDCs are transported through the water cycle, polluting the water and affecting the normal growth of aquatic organisms [41]. E-EDCs exposure in the environment is caused by the release and migration or the degradation of related compounds, which mostly occurs in the production and processing of antioxidants in food contact materials, epoxy resins, and polyvinyl chloride plastics. E-EDCs can enter the organism through dietary intake, respiration, and skin contact absorption, in which dietary intake is considered to be the main source of exposure and is more likely to be exposed than non-dietary sources [42]. E-EDCs may also be released into the environment through sewage discharge and nonpoint source runoff. Livestock feed and aquaculture are important reasons for the nonpoint source runoff components of e-EDCs. In addition, emissions of industrial and hospital wastes are also considered a major source of e-EDC contamination in the environment (Figure 2) [43]. In modern dairy production, approximately 75% of milk is produced from pregnant cows [44]. Endogenous natural estrogens produced by pregnant cows may enter milk through glucuronide acidification and sulfation metabolites. Glucuronide and sulfate metabolites are broken down by glucuronidase and sulfatase at different rates, bringing them back to their active free forms of estrogen [45]. To achieve rapid growth of the milking animals and increase milk production, various anabolic steroids and nonsteroidal synthetic estrogens (HEX, DES, DS, and BPA) are sometimes illegally applied as growth promoters [46]. When cows are fed, phytoestrogens entering the body can be decomposed into compounds without estrogen activity or excreted through feces and urine after passing through the rumen and intestine and may even be transferred to milk [47]. However, their activation occurs after the ingestion of β-glucuronidase by the human gastrointestinal tract [48]. In the process of milk production, BPA enters the milk chain at different time points through PVC tubes for milking and transfer from raw milk to storage tanks (Figure 3) [49]. Milk and dairy products are direct products of dairy cattle, which have a high chance of being contaminated by e-EDCs, endangering human health.

## 2. Sample Preparation of e-EDCs Detection Methods

The residual levels of e-EDCs in the milk matrix were quite low [50]. The matrix of milk and dairy products is rather complex, is rich in proteins and lipids, and should be precipitated and separated before extraction [51]. After protein removal, hydrolysis, extraction, cleaning, pre-concentration, and other processing steps, small molecules are retained, and the target analyte can be separated from the milk [52]. Therefore, it is essential to use appropriate sample pre-treatment methods before analysis and detection. Currently, the reported extraction approaches for e-EDCs include solid-phase extraction (SPE), solid-phase microextraction (SPME), magnetic solid-phase microextraction (MSPE), and liquid-phase microextraction (LPME) (Table 2).

### 2.1. SPE

SPE is the separation of isolates based on the partition coefficient between liquid and solid phases [83]. SPE devices include sorbent materials typically packed in columns and loaded with sample extracts to capture compounds of interest. Extraction loading is usually followed by one step to remove contaminants and then eluted with a small volume of different solvents to preconcentrate the analyte prior to instrumental analysis [84] or used as a pre-concentration and purification step at the same time [85]. One of the advantages of this method is that the analytes adsorbed on the SPE column are relatively stable and can be stored for a long time without changing their concentration or properties; however, SPE has poor selectivity and large solvent consumption [53]. Moreover, SPE pretreatments have a high recovery rate, few operational steps, and short analysis time [54]. Zhang et al. established an SPE method based on estrogen response elements to enrich and purify e-EDCs. First, triphenylamine was used to block the carboxylated silica gel coupled with the ER and an estrogen-responsive element. Blocking eliminates non-specific adsorption of unreacted carboxyl groups on silica gel, resulting in the specific binding of e-EDCs to the ER. The blocked silica gel coupled to the ER and estrogen responsive element was transferred to an empty SPE column and the e-EDCs standard solution was injected. This approach was coupled with high-performance liquid chromatography (HPLC) for the determination of BPA, 17β-E2, and DES in a liquid milk matrix, and the recovery rate of addition was 84.1–113.6%. The detection and quantification limits of this method were 1 × 10^−6^–5 × 10^−6^ mg·mL^−1^, respectively [55]. An automated on-line SPE-HPLC was developed to analyze E3, DES, and E1 in milk. First, samples were precipitated with acetonitrile and then purified and enriched on an online SPE column with polar-enhanced polymer. The supernatant was injected into an online SPE-HPLC system for detection. The recoveries were 70.82–112.90% [56].

### 2.2. SPME

SPME is a solvent-free technology widely applied in food, forensics, biomedical, and environmental fields [86]. The primary working principle of SPME is the mass transfer process, which forms a compound distribution balance between the sample matrix and coated adsorbent [87]. SPME requires exposing a small amount of the extracted phase dispersed on a solid support to sample matrices [88]. The working process consisted of two steps: (1) partition of the target between the coated adsorbent and the sample matrix and (2) desorption of the concentrated extract into a suitable mobile phase [89]. It integrates sampling, extraction, concentration, and injection processes [90]. SPME technology has the advantages of a short pretreatment time, and reduced consumption of organic solvents, no special equipment, low cost, and high sensitivity [57]. One of its disadvantages is the limited choice of the commercial stationary phase and fiber coating because of the high affinity between the stationary phase and the target analyte. Moreover, commercial fibers suffer from low thermal and chemical stability, lack of mechanical stability of fused silica fibers, coating stripping, and a short effective period [58]. Macromolecular substances in milk adsorbed onto SPME fibers may reduce their extraction rate and reusability. The covalent organic frameworks LZU1 and Nafion were used to coat the stainless-steel wires, and the direct immersion SPME protected by a dialysis membrane was used with gas chromatography-flame ionization detection (GC-FID) to detect trace E2 in milk samples, and the relative recovery was 77.27–108.26% [59]. Lan et al. developed a new automated SPME sampling method incorporating a magnetic molecularly imprinted polymer (MMIP) as a fiber coating for the quantitative enrichment of estrogen in milk powder. MMIP-SPME showed good sensitivity and binding to E1, E2, E3, and DES under optimized conditions, and their recoveries were 80.8–96.6%, 81.5–93.3%, 77.3–95.1%, and 79.4–92.2%, respectively [60].

### 2.3. MSPE

MSPE is a dispersive SPE that applies magnetic adsorbents to bind target analytes. First, the adsorbent is easily separated using an external magnetic field. Suitable solvents were then added to elute the analytes, followed by magnetic separation, and the liquid phase was collected for further analysis [91]. This method is suitable for different concentrations of organic and inorganic analytes and can be separated from complex matrices. It is a green, fast, and clean method with less time, without wasting energy or harmful organic solvents and has the characteristics of a high recovery rate and few extraction steps [61]. Furthermore, this method can use external magnets to quickly separate adsorbed targets from the sample solution without column packing issues [62], thereby avoiding column plugging and high-pressure limitations [63]. However, this method also has the difficulty and complexity of magnetic-material synthesis [64]. Scientists have established an MSPE method based on a graphitic carbon nitride for rapid and easy analysis of estrogen in milk powders. MSPE combined with HPLC showed that the linear ranges of enhancement factors for E2, 17α-EE2, E1, and HEX were 2–200, 1.5–150, and 3–300 µg·kg^−1^, respectively. The recovery was 75.1–97.2% [65]. Yang et al. used MSPE-HPLC to detect E2 in milk powders. In this study, novel mesoporous yolk–shell structure magnetic molecularly imprinted polymers (MYS-MMIPS) were established using organosilicon as the imprinting layer and E2 as the template. Using MYS-MMIPs as an adsorbent, E2 was quickly separated and enriched by MSPE with a recovery of 88.3–102.4% [66]. A novel sensitive method for the analysis of BPA, E2, and DES in milk was established by combining magnetic Fe_3_O_4_@MIL-53(Al) frameworks with HPLC-photodiode array detection. Fe_3_O_4_@MIL-53(Al) was used as the MSPE adsorbent to extract three e-EDCs from milk. The recovery rate of the method was 88.17–107.58% [67]. The magnetic molecularly imprinted polymer-assisted MSPE technique was applied for the rapid determination of three e-EDCs, nonylphenol, BPA, and HEX in milk samples. The average recovery rate of the three e-EDCs was 89.9–98.7% [68].

### 2.4. LPME

Currently, LPME is the most widely used sample extraction method. In a simple step, LPME can be used to extract, concentrate, and inject samples. It usually occurs between a few microliters of water-insoluble solvent and the aqueous phase containing analytes [92]. The liquid extraction phase of LPME was limited to a range of microliters. At present, LPME is divided into two methods: two-phase and three-phase LPME. Direct contact between the extraction phase and the sample solution is beneficial to the extraction process; however, the selectivity and sample removal are reduced, and the extraction solvent can only be used for water-immiscible organic liquids. The sample solution and the final acceptor phase are separated by a third solvent that is insoluble in the two phases in three-phase LPME, which significantly improves the selectivity of the method [93]. LPME can be extracted in different modes and can be divided into three categories: single-drop microextraction (SDME), dispersed LPME (DLLME), and hollow-fiber LPME (HF-LPME) [94]. The latter two have been used for milk and dairy product pretreatment.

#### 2.4.1. DLLME

DLLME has become an environmentally benign technique for sample preparation because of its simple operation, high speed, and low consumption of solvents and reagents [69]. DLLME is supported by the formation of ternary solvent systems, including aqueous solutions, water-immiscible solvents as extraction solvents, and dispersed solvents that are miscible with samples and extraction solvents. The contact of the three components formed a turbid solution containing many droplets, which maximized the interface between the phases and promoted the rapid distribution of analytes in the extraction solvent. The extraction solvent that enriches the analytes can then be separated from the rest of the system by centrifugation [95]. However, DLLME has high requirements for extraction and dispersed solvents. Extraction solvents should have higher density than water, and lower solubility in the aqueous phase. Dispersed solvents should have good miscibility with extraction solvents and the aqueous phase [70]. A rapid and inexpensive method based on DLLME sample processing was proposed to detect BPA, bisphenol F (BPF), Bisphenol S (BPS), parabens, and benzophenones in human milk, and the recovery values of all analytes were above 90.2% [71]. Feng et al. established a method for the efficient separation and determination of trace amounts of estrogens including E1, E2, chloromadinone 17-acetate (CMA), megestrol 17-acetate (MGA), 17 α-hydroxyprogesterone (HP), and medroxyprogesterone 17-acetate (MPA) in milk by combining DLLME with HPLC using magnetic ionic liquids as extraction solvent. The recovery rate was 98.5–109.3% [72].

#### 2.4.2. HF-LPME

HF-LPME devices are porous hollow fiber membranes in impregnated with organic solvents, which are based on the use of a supported liquid membrane (SLM) containing the extraction phase and separating it into the sample phase, resulting in a high specific surface area extraction and high enrichment factors [96]. The most important advantages of HF-LPME are its simple operation, low consumption of organic solvents, and low costs [73]. However, the limitation of HF-LPME is that it is only applicable to analytes whose functional groups are ionized in a specific pH range [74]. HF-LPME uses porous hollow fibers, which reduce or eliminate the problem of sample matrix generation and are divided into two main types: two-phase and three-phase. The HF-LPME format is highly flexible, and hollow fibers can be used in a U-shape or suspended from the needle tip [97]. The hollow fiber wall pores and cavities were filled with organic solvents that were insoluble in the sample aqueous solution in the two-phase HF-LPME. This method can be used for the extraction and enrichment of low-polarity compounds [98]. Compounds are extracted from the donor phase to the organic phase and then back extracted to the acceptor phase in the three-phase mode. Three-phase HF-LPME is suitable for acidic or basic hydrophobic compounds and can be used in conjunction with HPLC-tandem mass spectrometry (HPLC-MS/MS), liquid chromatography (LC), capillary electrophoresis (CE), and fluorescence detection [99]. Scientists have validated the potential of HF-LPME-based methods for the extraction of nine e-EDCs from different dairy products. The recoveries of E1, 17β-E2, 17α-E2, E3, 17α-EE2, DES, DS, HEX, and 2-hydroxyestradiol (2-OHE2) were all above 82% [75]. Wang et al. established a vortex assisted HF-LPME-HPLC method for the determination of 17β-E2, E1, and DES in milk samples. The recovery rates of this method in whole milk and skim milk samples were 86.24–94.25% [76]. LPME with a hollow fiber-based stirring extraction bar was used to extract E1, 17α-E2, 17β-E2, 17α-EE2, and E3 from milk. Stirring extraction bars are both stir bars for microextraction and extractors for analytes, enabling extraction, cleanup, and concentration in a single step. The stir extraction bar was easily separated from the extraction system using a magnet after the extraction was completed, and the recoveries were 93.6–104.6% [77].

### 2.5. The Quick, Easy, Cheap, Effective, Rugged and Safe (QuEChERS)

QuEChERS is a simple and direct extraction technology that includes initial segmentation, followed by extraction and cleaning by dispersed SPE [78]. The method includes extraction with acetonitrile and the addition of a salt mixture (MgSO_4_ and NaCl) for distribution purification. The addition of an appropriate adsorbent can also be purified using SPE, which has the advantages of low cost, simple operation, sensitive detection, and short time consumption [79]. However, the QuEChERS technique is highly dependent on the nature of the target analyte, substrate composition, experimental equipment, and temperature during operation [78]. Xiong et al. established a method for the simultaneous determination of nine bisphenols in milk samples by HPLC with a fluorescence detector. The samples were extracted using acetonitrile and cleaned using QuEChERS. The recoveries of the nine bisphenols in the spiked samples were 75.82–93.86% [80]. In another study, after extraction and clean-up of steroid hormones from raw milk (cow milk, goat milk, and buffalo milk) using a modified QuEChERS method, they were analyzed using ultra performance liquid chromatography-quadrupole time of flight mass spectrometry (UPLC-QTOF-MS). This method has a low detection limit and high recovery for most steroid hormones. The spiked recoveries of the matrix external standard method were 74.2–99.7% [81]. An improved method based on QuEChERS was proposed for the detection of 26 potential EDCs in milk, including E1, E2, E3, DES, BPA, and bisphenol B (BPB). At the experimental concentrations, the recoveries were 77.7–107.5% [82].

## 3. Detection Methods

Milk contains various proteins, fats, minerals, and carbohydrates, which may reduce detection sensitivity and even lead to detection failure [100]. E-EDCs in milk mainly exist as conjugated metabolites and biologically active free estrogens [101]. Classical estrogen detection mainly includes instrumental analysis methods such as gas chromatography-mass spectrometry (GC-MS), liquid chromatography-mass spectrometry (LC-MS), liquid chromatography-tandem mass spectrometry (LC-MS/MS), and biological analysis methods such as enzyme-linked immunosorbent assay (ELISA) (Table 3).

### 3.1. LC-MS

LC-MS is a combination of two selective techniques that allows the isolation and measurement of analytes of interest in highly complex mixtures, such as proteins or peptides, and simultaneously detects specific compounds according to their elemental and structural characteristics. LC distinguishes the physicochemical properties of compounds, whereas MS distinguishes the masses of compounds (especially their mass-to-charge ratios) [136,137]. BPA concentrations in milk, drinking water, and food samples were analyzed using LC-MS to assess the risk of passing through milk to the offspring [102]. The advantages of LC-MS/MS include its high selectivity and sensitivity, multi-analytical capability, and high throughput [138]. A variety of estrogens in human urine, serum, and breast milk can be determined using LC-MS/MS. The temperatures of the column chamber and autosampler were 40 °C and 10 °C, respectively. LC-MS grade water of eluent A (0.3 mM ammonium fluoride) and eluent B (acetonitrile) were used as the mobile phase. Multiple reaction monitoring (MRM) experiments were performed using rapid polarity switching in positive and negative electrospray ionization (ESI) modes. The limits of quantification (LOQ) were 0.015–5, 0.03–14, and 0.03–4.6 µg·L^−1^, respectively [103]. To evaluate the endogenous steroid hormones in isolated colostrum and colostrum powder, a sensitive LC-MS/MS method was established. An Acquity HSS T3 column (1.8 μm, 2.1 mm × 100 mm) connected to the VanGuard front column (1.8 μm) was used for chromatographic separation. Eluent A and eluent B were 0.007% formic acid water and methanol supplemented with 0.007% formic acid, respectively. The column oven temperature was 60 °C and estrogens were monitored by tandem MS using negative ESI mode. E1, 17α-E2, and 17β-E2 were also detected [104]. A similar method was developed to quantify six sex hormones (pregnenolone, progesterone, E1, testosterone, androstenedione, and dehydroepiandrosterone) in the milk. The pretreated samples were added to the analytical instrument, and the hormone concentration was measured at ng·dL^−1^. Its application in real raw milk samples statistically confirmed the difference in milk between pregnant and nonpregnant cows [105]. Another study quantified 12 hormones at the level of ng·kg^−1^ in milk using LC-MS/MS. This technique used a C18 column (2.1 mm × 100 mm, 1.7 µm). Mobile phase A and mobile phase B of LC were water and methanol containing 0.1% formic acid, respectively, and the column oven temperature was 40 °C. MS used the positive ESI mode. The levels of 17α-E2, 17β-E2, and E1 were 31, 6, and 159 ng·kg^−1^, respectively [106]. The combination of SPE and LC-MS/MS was used to extract E1, 17β-E2, E3, 17α-EE2, and conjugated estrogen metabolites. The mobile phases A and B were 5 mM ammonium hydroxide/methanol (96/4, *v*/*v*) and 5 mM ammonium hydroxide/methanol/acetonitrile (10/10/80, *v*/*v*/*v*). Ionization was achieved through negative ESI mode. The LODs of free e-EDCs in milk were 6, 10, 10, and 37 ng·L^−1^, respectively. Several additional conjugated estrogenic metabolites were found at 2 ng·L^−1^ for estrone-3-sulfate, 7 ng·L^−1^ for estrone-3-glucuronide, 6 ng·L^−1^ for 17β-estradiol-3,17β-sulfate disodium salt, and 7 ng·L^−1^ for 17α-ethinylestradiol-3-glucuronide. A peak with a nominal mass of molecular ions similar to sulfated E2 ([M-H]^−^, *m*/*z* 351) was also found during samples analysis. This study used QTOF-MS and LC connected to a LCQ Advantage ion trap (LC–IT-MS) were used to demonstrate that the unknown peak was not an E2-3S isomer [45]. Socas-Rodríguez et al. developed a method for monitoring milk and dairy products for the presence of various estrogenic substances. This method requires enzymatic hydrolysis followed by QuEChERS-based extraction. Then, ultrahigh-performance liquid chromatography (UHPLC) combined with MS/MS was used for analysis. An Acquity UPLC BEH C18 column (100 mm × 2.1 mm, 1.7 μm, Waters) was used for the technology. Mobile phases A and B were 2 mM NH_4_OH and MeOH:MeCN (50:50, *v*/*v*), respectively, and the column oven temperature was 40 °C. MRM was performed in negative ESI mode. The LOQs were 0.02–0.60 µg·L^−1^ and 0.02–0.90 µg·kg^−1^, respectively [5]. A method for the selective determination of seven estrogens in milk samples via ultra-high performance liquid chromatography-tandem mass spectrometry (UHPLC-MS/MS) was also established. LC used Kinetex F5 column (2.1 × 100 mm, 2.6 µm) for material separation. Column oven temperature of LC was 40 °C, mobile phase A was water, and mobile phase B was methanol. MS systems with heated ESI sources used negative ionization modes. The limit of detection (LOD) of this method was 0.10–0.35 µg·L^−1^ [50]. Huang et al. used UHPLC-MS/MS to simultaneously determine of aflatoxin M1, ochratoxin A, ZEN, and α-zearalenol in milk. Solvent A (methanol) and solvent B (0.1% (*v*/*v*) ammonia) were used for LC using an UHPLC BEH C18 column (1.7 μm, 50 mm × 2.1 mm). In addition, column and sample temperatures were 40 °C and 25 °C, respectively. Analysis was performed in negative ESI mode with higher sensitivity and stability. The LOQ of the method was 0.003–0.015 µg·kg^−1^ [107].

### 3.2. GC-MS

GC-MS is widely used, because it has good selectivity, high separation degree, high sensitivity and repeatability, and relatively stable in the process of use. The two ionization forms of GC-MS are electron ionization (EI) and chemical ionization (CI). In addition, GC-MS is easy to use and provides insight into the identification of compounds [139]. Scientists have established a method for determining 23 different EDCs in dairy products. The samples were adjusted by the addition of acetonitrile, centrifugation, and cleanup of the extract using sequential SPE. The EDCs in the extracts derived by microwave heating were quantified using GC-MS. The method was barely affected by matrix effects, and the LOD of BPA was 6–40 ng·kg^−1^ [108]. A similar method was developed for the analysis of free and total BPA in human milk samples, and the detection concentration reached ng·g^−1^ [109]. The major steroid hormones in milk and eggs have been studied, and an analytical method based on GC-MS/MS has been developed to measure ultra-trace levels of steroids in foods. The LODs of estrogen in milk and eggs were 5 and 30 ng·kg^−1^, respectively [110]. An environmentally friendly method based on GC-MS/MS was developed to extract and determine the selected estrogenic compounds (including 17α-E2, 17β-E2, and 17α-EE2) from whole dairy cows, semi-skimmed goat milk, and all-natural yogurt. The LODs of whole milk and semi-skimmed goat milk were 0.06–2.55 and 0.04–1.70 µg·L^−1^, and the LOQs were 0.16–6.11 and 0.11–4.86 µg·L^−1^, respectively. The LOD and LOQ of natural yogurt were 0.07–3.73 and 0.23–9.81 µg·L^−1^, respectively [111].

### 3.3. Biosensors for Estrogens

At present, biosensors are widely used in the fields of biomedical diagnosis, immediate monitoring of therapeutic and disease progress, food control, drug discovery, forensics, and biomedical research, and are also regarded as analytical tools [140]. Biosensors, devices that use biochemical reactions to analyze chemicals, are innovative high-efficiency solutions that are specific to the target analyte and offer high precision in complex matrices [141]. Biosensors generally consist of a molecular recognition layer, sensor, and signal generator [142]. The molecular recognition layer is composed of biometric elements, such as enzymes, receptor proteins, probe molecules, and cell receptors fixed on the surface of the transducer, which react with the target biomolecules and convert biological interactions into physical signals [143]. Because of their combination with high-affinity biomolecules, analyte sensitivity and selectivity can be achieved [144]. Biosensors have become increasingly popular for food estrogens detection because of their high selectivity and sensitivity, rapid response, low cost, continuous online monitoring in complex systems, stability, repeatability, high automation, miniaturization, and integration [145]. Electrochemical biosensors, optical biosensors, and photoelectrochemical biosensors for the detection of estrogen in food are described [146].

#### 3.3.1. Electrochemical Biosensors

Pan et al. used an indirect competitive immunoassay to prepare an electrochemical immunosensor that can simultaneously detect four phenolic estrogens: HEX, DES, DS, and BPA. Differential pulse voltammetry was applied and the amperometric response sequence was EDS > DS > BPA > HEX, and the LODs were 0.25, 0.15, 0.20, and 0.25 ng·mL^−1^, respectively. This method was validated using milk powder, and the results displayed high accuracy [112]. A simple electrochemical sensor using the split DNA aptamer as the recognition agent was presented to determine 17β-E2. The split aptamer binds to 17β-E2 and establishes a complex as a bridge on the electrode surface, enabling ultrasensitive detection. This aptasensor can recognize 17β-E2 within 30 min without requiring complex procedures or expensive equipment. The LOD of 17β-E2 in milk samples using this method was 0.7 pM [113]. A fluorescent aptamer with high sensitivity and simple operation was designed to detect E2 by hybridization chain reaction (HCR) and horseradish peroxidase (HRP) amplification. The complementary strand (cmDNA) competes with the E2 to E2 aptamer modified on magnetic beads so that the unbound cmDNA is collected and captured by the polystyrene microspheres, inducing HCR to produce a large number of biotinylated sites. Owing to the excellent catalytic performance of streptavidin-horseradish peroxidase, a high-sensitivity fluorescence signal was obtained at low levels of E2. The linear range of E2 detection was 1–100 pg·mL^−1^, and the LOD is 0.2 pg·mL^−1^, which show good adaptability in the milk sample [114]. A novel E2 fluorescent aptamer sensor was developed using E2 aptamer-labeled carbon quantum dots and complementary DNA-modified Fe_3_O_4_ nanoparticles. Under the optimal conditions, the linear range of the sensor pair E2 was 10^−11^–10^−6^ M, and the LOD was 3.48 × 10^−12^ M. The sensor showed good selectivity and repeatability for the analysis of E2 in milk [115]. Au nanoparticles coated with boron-doped diamond and a 6-mercapto-1-hexanol aptamer sensor were designed and fabricated. The sensor could detect trace BPA with good linearity in the range of 1.0 × 10^−14^–1.0 × 10^−9^ mol·L^−1^, and the LOD was 7.2 × 10^−15^ mol·L^−1^ [116]. An unlabeled electrochemical sensor for BPA detection based on gold nanoparticles-dotted graphene was developed. The linear concentration range of BPA was 0.01–10 µM, and the LOD is 5 µM. The sensor was successfully applied to determine BPA in liquid milk and milk powder [117]. Karthika et al. developed a molecularly imprinted polymer sensor for BPA analysis; the required film was a polypyrrole-based imprinted polymer film synthesized by electrochemical polymerization on electrochemically reduced graphene oxide. BPA concentrations showed a linear relationship between 750 and 0.5 mol·L^−1^ with a LOD of 0.2 nmol·L^−1^. The results from spiked milk showed good recovery and repeatability [118].

#### 3.3.2. Optical Biosensors

A biosensor based on total internal reflection fluorescence was developed to detect progesterone in milk samples. The LOD of progesterone in UHT milk, fresh milk, and raw milk by the method was 45.5–56.1 pg·mL^−1^, which is lower than the level of progesterone in commercial milk and randomly purchased raw milk [119]. Daems et al. developed a biosensor based on automated fiber-optic surface plasmon resonance to determine progesterone in milk. The LOD detected by this biosensor on milk with additive was 0.5 ng·mL^−1^. These results are consistent with those of ELISA [120]. A novel surface plasmon resonance biosensor chip relying on magnetic nanoparticles was used to detect progesterone in milk. The sensitivity for milk was 0.038 ng·mL^−1^ [121]. A novel biosensor based on poly(N-isopropylacrylamide) microgel was used to determine E2 using E2 combined with a 75-mer DNA aptamer. The aptamer first binds to E2 and the product undergoes a conformational change that prevents the diffusion of salt ions into the microgel layer, which is proportional to the E2 level. The LOD of nonfat milk, 2% milk, and farm milk determined by this method was 0.9, 8.4, and 4–9.1 pg·mL^−1^, respectively [122]. Ren et al. developed an up-conversion fluorescent aptamer sensor based on black phosphorus nanohybridization and a self-assembled DNA tetrahedral double amplification strategy for the rapid determination of BPA and E2. The linear ranges of BPA and E2 were 0.01–100 and 0.1–100 ng·mL^−1^, and their LODs were 7.8 and 92 pg·mL^−1^, respectively. The capture time was reduced to 10 min [123]. An indirect probe-based independent component analysis method was used for the highly sensitive detection of E2. Its lowest visible LOD was 0.2 ng·mL^−1^, and it is successfully applied to detect E2 in milk samples [124]. An immuno-filter paper strip based on the photothermal effect of black phosphorus nanoplate was proposed for the detection of E2. Temperature change was used in the measurement instead of the traditional color change during signal reading. The LOD of E2 was as low as 0.104 ng·mL^−1^ [125]. A BPA fluorescence biosensor based on a DNA amplification circuit and a Mg^2+^ dependent DNAzyme was developed. The sensor employed an anti-BPA aptamer as the recognition element for BPA binding. The double-labeled substrate DNA was cleaved into two segments by synergistic DNA hybridization to form an Mg^2+^-dependent catalytic DNAzyme. The separation of the fluorophore and quencher resulted in a high fluorescence response to BPA determination, and the LOD was 50 fM in milk samples [126].

#### 3.3.3. Photoelectrochemical Biosensors

Photoelectrochemical sensing technology has attracted increasing attention in the field of biological and environmental molecular detection owing to its high sensitivity, low cost, and ease of miniaturization. Quantitative detection is typically based on photoelectron transfer at the electrode/electrolyte interface and photocurrent changes due to the interaction between the target and the photoactive matrix or probe [147]. A highly sensitive photoelectrochemical sensor based on Au nanorods was developed for the detection of trace E2. The prepared photoelectrochemical sensor has good analytical performance for E2 in the range of 1 × 10^−15^ to 1 × 10^−9^ M under optimal conditions, and the LOD was 3.3 × 10^−16^ M [127]. Qiao et al. established a BPA unlabeled photoelectrochemical sensor based on gold nanoparticles sensitized ZnO nanoporous surface plasmon resonance. The LOD of the method was 0.5 nmol·L^−1^, and the BPA in liquid milk samples has been successfully detected [128].

### 3.4. ELISA

ELISA is a quantitative assay that displays antigen-antibody responses by color changes of enzyme-linked conjugants and enzyme substrates and is used to determine the presence and concentration of target molecules [148]. Bai et al. prepared polyclonal and monoclonal antibodies and established an indirect competitive ELISA to detect 17β-E2 in milk. The LOD was 0.093 µg·L^−1^, with sufficient sensitivity for the detection of 17β-E2 in milk [129]. A plasmonic biomimetic ELISA method using a molecularly imprinted polymer membrane was developed for the ultrasensitive and on-site visual determination of BPA. With an increase in BPA concentration, the color of the samples changed significantly. Plasmonic biomimetic ELISA is highly sensitive, cost-effective, and simple to perform. This method can detect BPA with naked-eye observations and a visual LOD of 40 pg·mL^−1^. In quantitative analysis, the proposed method showed a good dynamic linear response at logarithmic concentrations [130]. Samiee et al. used ELISA to determine aflatoxin M1, ochratoxin A, and ZEN in human milk. Among them, the concentration of aflatoxin M1 was 5.98 ng·L^−1^, and the levels of ochratoxin A and ZEN were lower than the LOD (<5 ng·L^−1^) [131]. Kuruto-niwa et al. measured BPA concentrations in human colostrum by ELISA and detected BPA in 101 samples in a concentration range of 1–7 ng·mL^−1^. The mean concentration was 3.41 ± 0.13 (mean ± SD) ng·mL^−1^ [132].

### 3.5. Surface-Enhanced Raman Spectroscopy (SERS)

Raman spectroscopy is rarely used in the study of estrogens, mainly because of its inherently weak signals and fluorescence interference caused by fluorescent pollutants in most samples. Special Raman techniques such as SERS are typically required [149]. SERS is a highly sensitive analytical technique for detecting single molecules [150]. It has been used to discriminate and quantify E1, E2, E3, and 17α-EE2. The simulated Raman spectra techniques are correlated with the experimental data to identify unique marker peaks, which are useful for distinguishing different estrogen molecules [151]. SERS based on gold nanoparticles modified with Zn^2+^ as polymerization agents was established to detect BPA residues in milk with high sensitivity. Under optimal conditions, the LOD of BPA was 4.3 × 10^−9^ moL·L^−1^ (0.98 × 10^−3^ mg·kg^−1^), lower than the European Standard (0.6 mg·kg^−1^) [133]. An easy and fast method based on surface-enhanced resonance Raman scattering (SERRS) was established for the detection of trace amounts of phenolic estrogens. The azo compounds showed strong Raman spectroscopy activity owing to the coupling reaction. The LOD of E1, E2, and BPA were as low as 0.2 × 10^−4^ M. Each Raman spectroscopy fingerprint of azo dye is a specific hormone. This technique can be used not only for the detection of phenolic hormones in coupling reactions, but also in the study of other phenolic molecules [134]. A simple and rapid method based on SERRS for the detection of trace phenolic estrogens was developed. The results showed that SERRS has high selectivity for azo dyes, even in complex mixtures. The LOD of the method was approximately 0.1 µg·kg^−1^ [135].

## 4. Conclusions and Future Perspectives

At present, milk and dairy products are common foods in people’s daily lives; however, they are also a major source of estrogen disruptors. E-EDCs have become a global health problem, posing a threat to human health. Trace e-EDCs enter the human body and cause potential harm through accumulation, including damage to the endocrine system. Therefore, fast, sensitive, and efficient methods are needed to detect multiple low-dose and ultra-low-dose e-EDCs. In this review, the principles of LC-MS/MS, GC-MS, estrogen biosensors, ELISA, Raman spectroscopy, and other detection methods are introduced, and sample pretreatment is summarized. In this regard, many techniques can reach the detection level of ng·L^−1^ by improving the pre-processing steps or innovative detection methods. The current pre-processing and detection technologies that are applied in milk and dairy products are either expensive or complicated to operate. HPLC and GC not only need professional operators, but also cannot be used for on-site testing. Electrochemical sensors and SERS require too much time during sample preparation and detection and are not widely used in commercial detection. SPE in pretreatment methods and ELISA, LC-MS/MS, and GC-MS in detection methods are widely used. In addition, real-time monitoring should be performed on how e-EDCs enter raw milk, and the detection instruments and conditions can be optimized to make them portable and green and improve the detection efficiency. Based on the above problems, the development of portable, automatic, and low-cost pretreatment and detection technology should be strengthened in the future.

## Figures and Tables

**Figure 1 foods-11-03057-f001:**
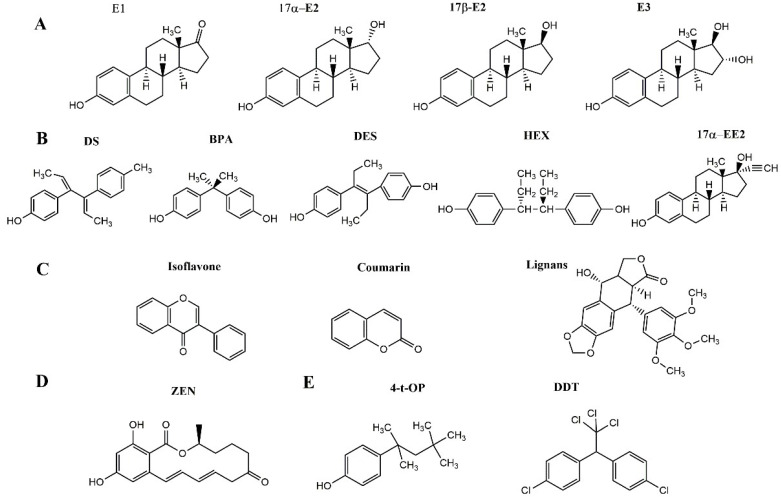
Chemical structures of different kinds of e-EDCs. (**A**): Endogenous natural estrogens. E1, estrone; 17α-E2, 17α-estradiol; 17β-E2, 17β-estradiol; E3, estriol. (**B**): Synthetic estrogens. DS, dienestrol; BPA, bisphenol A; DES, diethylstilbestrol; HEX, hexestrol; 17α-EE2,17α-ethynyl estradiol. (**C**): Phytoestrogens. Isoflavones; Coumarins; Lignans. (**D**): Estrogen-active mycotoxins. ZEN, zearalenone. (**E**): Alkylphenols. 4-t-OP, 4-tert-octylphenol. Other chemicals. DDT, dichlorodiphenyltrichloroethane.

**Figure 2 foods-11-03057-f002:**
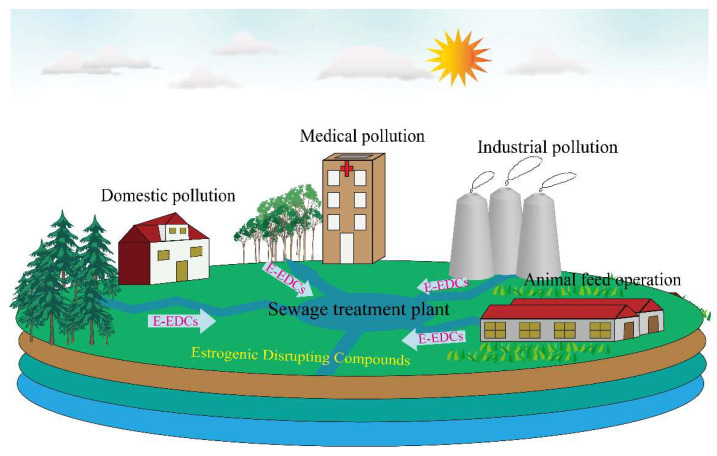
Sources of estrogen disrupting chemicals in environment.

**Figure 3 foods-11-03057-f003:**
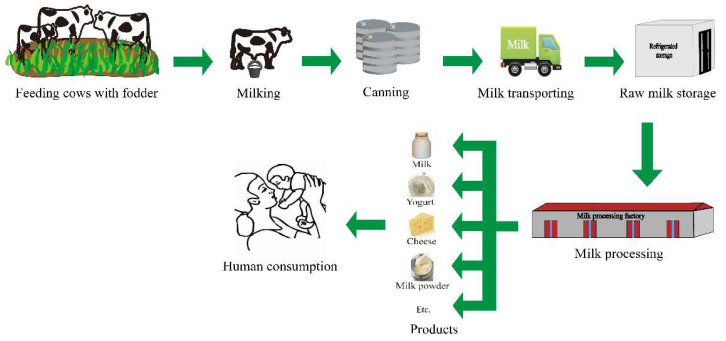
Sources of BPA in milk and dairy products.

**Table 2 foods-11-03057-t002:** Sample pretreatment methods of e-EDCs in milk and dairy products.

Technology	Strengths	Limitations	Samples	E-EDCs	Recovery	Refs.
SPE	High recovery rate; Few operation steps; Short analysis time; Analyte stabilization.	Poor selectivity; Large solvent consumption.	Milk	BPA, E2, DES	84.1 ± 8.2–113.6 ± 2.9%	[53,54,55,56]
			Milk	E3, DES, E1	70.82–112.90%	
SPME	Short pretreatment time; Less organic solvent consumption; No need for special equipment; Low cost; High sensitivity.	Limited options for commercial stationary phases and fibrous coatings; Low thermal and chemical stability.	Milk	E2	77.27–108.26%	[57,58,59,60]
			Milk powders	E1, E2, E3, DES	80.8–96.6%; 81.5–93.3%; 77.3–95.1%; 79.4–92.2%	
MSPE	Fast; Clean; Short time; High recovery; Few steps; Little waste; No column blockage problem.	Difficult and complex synthesis of magnetic materials.	Milk powders	E2,17α-EE2, E1, HEX	75.1–97.2%;	[61,62,63,64,65,66,67,68]
			Milk powders	E2	88.3–102.4%	
			Milk	BPA, E2, DES	88.17–107.58%	
			Milk samples	Nonylphenol, BPA, and HEX	89.9–98.7%	
DLLME	Simple operation; Fast speed; Low consumption of solvents and reagents.	DLLME has higher requirements for extraction and dispersed solvents.	Human milk	BPA, BPF, BPS parabens, and benzophenones	Above 90.2%	[69,70,71,72]
			Milk	E1, E2, CMA, MGA, HP, MPA	98.5–109.3%	
HF-LPME	Simple operation; Low consumption of organic solvents; Low cost.	Within a specific pH range.	Different dairy products	E1, 17β-E2, 17α-E2, E3, 17α-EE2, DES, DS, HEX, 2-OHE2	Above 82%	[73,74,75,76,77]
			Whole milk and skim milk	E2, E1, DES	98.5–109.3%	
			Milk	E1, 17α-E2, 17β-E2, 17α-EE2, and E3	93.6–104.6%	
QuEChERS	Low cost; Simple operation; Sensitive detection; Short time consumption.	Highly dependent on the nature of the target analyte, substrate composition, experimental equipment, and temperature.	Milk	Nine bisphenols	75.82–93.86%	[78,79,80,81,82]
			Raw milk	17β-E2, E3, E1, DES, progesterone	74.2–99.7%	
			Milk	E1, E2, E3, DES, BPA, and BPB	77.7–107.5%	

**Table 3 foods-11-03057-t003:** Methods for determination of e-EDCs in milk and dairy products.

Technology	Strengths	Sample	E-EDCs	LOD (s) or LOQ (s)	Refs.
LC-MS	High selectivity and sensitivity.	Milk	BPA	ppm levels	[102]
LC-MS/MS	Analytical capability; High throughput.	Breast milk	Xenoestrogens	0.03–4.6 µg·L^−1^	[103]
		Isolated colostrum and colostrum powder	E1, 17α-E2, 17β-E2;	E1 (5.51µg·L^−1^; 15 µg·kg^−1^), 17α-E2 (2.66 µg·L^−1^; 7.5 µg·kg^−1^) and 17β-E2 (2.28 µg·L^−1^; 3.3 µg·kg^−1^)	[104]
		Milk	Progesterone, E1	ng·dL^−1^ level	[105]
		Milk	17α-E2, 17β-E2, E1	31 ng·kg^−1^, 6 ng·kg^−1^, 159 ng·kg^−1^	[106]
		Milk	E1, 17β-E2, E3, 17α-EE2, and conjugated estrogen metabolites	ng·L^−1^ level	[45]
UHPLC-MS/MS	N.M.	Milk and yogurt	Various estrogenic compounds	0.02–0.60 µg·L^−1^, 0.02–0.90 µg·kg^−1^	[5]
		Milk	E1, 17β-E2, 17α-E2, E3, 17α-EE2, DES, HEX, DS	0.10–0.35 µg·L^−1^	[50]
		Milk	ZEN, and α-zearalenol	0.003–0.015 µg·kg^−1^	[107]
GC-MS	Good selectivity; high separation degree; High sensitivity; High repeatability; Relatively stable.	Dairy products	BPA;	6–40 ng·kg^−1^;	[108]
		Human milk	Free and total BPA	ng·g^−1^ level	[109]
		Milk	E1, 17β-E2, 17α-E2	5 ng·kg^−1^	[110]
		Different kinds of dairy products	17α-E2, 17β-E2, and 17α-EE2	µg·L^−1^ level	[111]
Electrochemical biosensors	N.M.	Milk powder	EDS, DS, BPA, HEX	0.25, 0.15, 0.20 and 0.25 ng·mL^−1^	[112]
		Milk	17β-E2	0.7 pM	[113]
		Milk	E2	0.2 pg·mL^−1^	[114]
		Milk	E2	3.48 × 10^−12^ M	[115]
		Milk	BPA	7.2 × 10^−15^ mol·L^−1^	[116]
		Liquid milk and milk powder	BPA	5 µM	[117]
		Bovine milk	BPA	0.2 nmol·L^−1^	[118]
Optical biosensors	N.M.	UHT milk, fresh milk and raw milk	Progesterone	45.5–56.1 pg·mL^−1^	[119]
		Milk	Progesterone	0.5 ng·mL^−1^	[120]
		Milk	Progesterone	0.038 ng·mL^−1^	[121]
		Nonfat milk, 2% milk, and farm milk	E2	0.9 pg·mL^−1^, 8.4 pg·mL^−1^, and 4–9.1 pg·mL^−1^	[122]
		Milk	BPA and E2	7.8 pg·mL^−1^ and 92 pg·mL^−1^	[123]
		Milk	E2	0.2 ng·mL^−1^	[124]
		Milk	E2	0.104 ng·mL^−1^	[125]
		Milk	BPA	50 fM	[126]
Photoelectrochemical biosensors	High sensitivity; Low cost; Easy miniaturization.	Milk powder	E2	3.3 × 10^−16^ M	[127]
		Liquid milk	BPA	0.5 nmol·L^−1^	[128]
ELISA	Highly sensitive; Cost-effective; Simple to perform.	Milk	17β-E2	0.093 µg·L^−1^	[129]
		Milk	BPA	40 pg·mL^−1^	[130]
		Human milk	ZEN	5 ng·L^−1^	[131]
		Human colostrum	BPA	ng·mL^−1^	[132]
SERS	Highly sensitive.	Milk	BPA	4.3 × 10^−9^ moL·L^−1^	[133]
		Infant formula	E1, E2 and BPA	0.2 × 10^−4^ M	[134]
		Infant formula	E2	0.1 µg·kg^−1^	[135]

LOD, limit of detection; LOQ, limit of quantification; N.M., not mentioned; ppm, parts per million.

## Data Availability

No new data were created or analyzed in this study. Data sharing is not applicable to this article.

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
