# Peer review of "Research Advances in the Analysis of Estrogenic Endocrine Disrupting Compounds in Milk and Dairy Products"

_foods, 2022, doi:10.3390/foods11193057_

Round 1

Reviewer 1 Report

The manuscript is very complete and describes different “traditional” and “new” methods to detect estrogenic endocrine disrupting compounds in milk and dairy products. In general, the manuscript is well designed although some minor remarks must be taken in consideration.

Minor remarks:

In my opinion, “detection” as a keyword is too general, I suggest changing this keyword.

The authors decided to evaluate the presence of these compounds in milk and dairy products, however, these compounds can be present in other foodstuffs of animal origin, as shown in table 3, moreover, in line 44 the authors say “Milk and dairy products are the sources of e-EDCs”. In my opinion, it must be indicated that milk and dairy products are a source of e-EDCs but not the only ones, as they can be present in other food matrices.

Line 53: Figure. 1 à There is an additional dot between the word and the number

In Table 1, in the MRL column, some information corresponds to MRL data but other corresponds to tolerable daily intakes, this information must be separated, or it must be indicated this difference, as they are not the same

 Line 92: estrogen receptors must be written abbreviated

Line 95: Fusarium spp. must be written in italics

Line 353: Limits of quantification (LOQ), must be abbreviated here as is the first time that appears in the text, before line 358

In sections 3.3.3. Photoelectrochemical biosensors and 3.4. ELISA given information is scarce compared to other methods, is it possible to add some information in these sections? The same for sections 4.3 and 4.4

Line 507: “and the LOQs are” à change are into were

Reviewer 2 Report

This manuscript described several techniques used in the pretreatment and detection of estrogenic endocrine disrupting compounds in milk and dairy products. The document structure is correct. However, I have several comments for the whole paper, which must be taken into account to improve the article before it will be considered for Foods.

-          Wording: authors should check the punctuation of the text, as some paragraphs should be divided.

-          Authors should always use the original reference. For example, the Codex Alimentarius Commission (CAC) and the Chinese National Food Safety Standard instead of reference 12, the European Union and the CAC instead of reference 13, the Ministry of Agriculture of China instead of reference 15, EPA recommendation instead of reference 17, reference 21 in line 100, etc.

-          Missing the reference of the phrase from lines 82-83.

-          Authors should update the limit for BPA to 0.05 mg/kg (Regulation 2018/213).

-          Authors should complete Table 1 with a column indicating the references.

-          The name of the species is written in italics (line 95).

-          This review focuses on milk and dairy products, perhaps delete lines 131-132 referring to drinking water, as well as lines 135-138 referring to zebrafish, etc.

-          In section 1.3, authors should include the possible routes of exposure (inhalation, ingestion, skin), as well as that BPA can be released from food packaging (lines 160-162).

-          Table 2: the table does not provide more information than that of the text, therefore, consider expanding it with more examples or eliminating it. Contains errors such as "Dimple" instead of "Simple", "17α-EE2" instead of "17α-E2". Explain the abbreviation for 2-OHE2. Information regarding the limitations of some techniques is lacking, as well as specifying the bisphenols analyzed with DLLME.

-          With some techniques, the authors include some descriptions of it, but few practical examples. For example, for SPE, DLLME, HF-LPME only one example is included.

-          The first time that an acronym is mentioned, it should be fully written. For example, MS/MS (line 293), QuEChERS (line 297), QTOF (line 312) etc.

-          Table 3: missing references, and the meaning of “N.M.” at the foot of the table. Many non-milk and dairy products are included in this table, which makes no sense.

-          Table 4: missing the meaning of “N.M.” at the foot of the table; the meaning of ppm, LOD and LOQ; and contains errors such as “7β-E2” instead of “17β-E2”.

-          I recommend merging sections 3.1, 3.1.1 and 3.1.2. into the same section and include more practical examples with data such as the column, mobile phase, ionization source used, etc.

-          In the case of GC-MS and ELISA, a brief description of the techniques is not included as in the other sections.

-          Please, check if it is correct to name “the microgel etalon of poly” in line 433, it sounds strange.

-          The information in lines 490-491 is repeated on line 494.

-          Assess whether it is necessary to include section 4 since, in reality, in all the previously mentioned works, both preprocessing and detection techniques are combined. Evaluate the possibility of including these examples in the tables.

-          Section 4.2: it may be interesting to include the m/z of the conjugated estrogenic metabolites.

Round 2

Reviewer 2 Report

The work has been remarkably improved.